# SYSCAPS: LANGUAGE INTERFACES FOR SIMULATION SURROGATES OF COMPLEX SYSTEMS

**Patrick Emami, Saumya Sinha,**[*] **Truc Nguyen**[*]
National Renewable Energy Lab
{Patrick.Emami,Saumya.Sinha,Truc.Nguyen}@nrel.gov

**Zhaonan Li**[*†]
Arizona State University
zhaonan2@asu.edu

## ABSTRACT

Surrogate models are used to predict the behavior of complex energy systems that are too expensive to simulate with traditional numerical methods. Our work introduces the use of language descriptions, which we call "system captions" or SysCaps, to interface with such surrogates. We argue that interacting with surrogates through text, particularly natural language, makes these models more accessible for both experts and non-experts. We introduce a lightweight multimodal text and timeseries regression model and a training pipeline that uses large language models (LLMs) to synthesize high-quality captions from simulation metadata. Our experiments on two real-world simulators of buildings and wind farms show that our SysCaps-augmented surrogates have better accuracy on held-out systems than traditional methods while enjoying new generalization abilities, such as handling semantically related descriptions of the same test system. Additional experiments also highlight the potential of SysCaps to unlock language-driven design space exploration and to regularize training through prompt augmentation.

## 1 INTRODUCTION

Data-driven surrogates enable computational scientists to efficiently predict the results of expensive numerical simulations that run on supercomputers (Lavin et al., 2021; Carter et al., 2023). Surrogates are particularly valuable for emulating simulations of *complex energy systems* (CES), which model dynamic interactions between humans, earth systems, and infrastructure. Examples of CES include buildings (Vazquez-Canteli et al., 2019; Dai et al., 2023; Bhavsar et al., 2023), electric vehicle fleets (Vepsäläinen et al., 2019), and microgrids (Du & Li, 2019). Advancing the science of CES contributes to efforts aimed at reducing emissions and improving the resiliency of power systems.

These surrogates perform a fairly standard regression task, predicting simulation output quantities of interest from *a*) an input system configuration and *b*) a deployment scenario. For example, we might want to predict the amount of energy a building will consume given *a*) a list of building characteristics and *b*) a weather timeseries spanning an entire year. In this case, this involves performing long sequence timeseries regression, which traditional regression techniques such as gradient-boosted decision trees have difficulty with (Bhavsar et al., 2023; Zhang et al., 2021).

Surrogate models are not only used by experts. Surrogates are also used to inform highly consequential policy and investment decisions about complex systems made by non-experts in industry and governments (Rackauckas & Abdelrehim, 2024), such as when planning to build and deploy a new renewable energy system (Harrison-Atlas et al., 2024). In this work, we design and analyze *language interfaces* for such surrogates. Intuitively, language interfaces make surrogate models more accessible, particularly for non-experts, by simplifying how we inspect and alter a complex system's configuration. Language interfaces are powerful—they ground interactions between humans and

---

[*]Equal contribution.
[†]Work done during internship at NREL.

machines in the human's preferred way (Vaithilingam et al., 2024). The idea of using language to create interfaces for complex data or models is not new (Hendrix et al., 1978; Quamar et al., 2022), but interest has renewed due to the success of large language models (LLMs) and their demonstrated ability to generate high-quality synthetic natural captions (Schick & Schütze, 2021; Doh et al., 2023; Mei et al., 2023a; Hegselmann et al., 2023). Our work defines a "system caption", or **SysCap**, as text-based descriptions of *knowledge about the system* being simulated. The only available knowledge our work assumes is the system configuration found in simulator metadata files as lists of attributes.

In general, it is unknown whether textual inputs, and particularly *natural language* inputs, are suitable for real-world *tabular* regression tasks. Tabular data, such as the system attributes in question, are sets of both discrete (categorical, binary, or string) and continuous (numeric) variables. Previous work demonstrated inconclusive evidence when using language models to do tabular regression from text-encoded inputs, with and without modifications to the architecture (Dinh et al., 2022; Jablonka et al., 2024; Bellamy et al., 2023; Yan et al., 2024), motivating further study. Regression with text-encoded tabular inputs is promising because *a*) language is a more intuitive and flexible user interface than traditional encoding strategies (e.g., one-hot encodings), and *b*) using language embeddings to encode system attributes unlocks the ability to exploit the semantic information contained in SysCaps to generalize across related systems.

Our paper introduces a framework for training *multimodal* surrogates for CES with text (for system attributes) and timeseries inputs (for the deployment scenario) and makes contributions towards addressing the following technical challenges:

- We introduce a simple and lightweight multimodal surrogate model architecture for time-series regression that *a*) fuses text embeddings obtained from fine-tuned language models (LMs) with *b*) timeseries encoded by a bidirectional sequence encoder. We expect this to be insightful for future multimodal text and timeseries studies.

- To address the lack of human-labeled natural language descriptions of complex systems, we describe a process that uses LLMs to generate high-quality natural language SysCaps from simulation metadata. Although LLMs have previously been used to generate text captions from metadata (Doh et al., 2023; Mei et al., 2023b), we believe our application to multi-modal surrogate modeling is novel.

- We develop an automatic evaluation strategy to assess caption quality–specifically, we estimate the rate at which ground truth attributes appear in the synthetic description with a multiclass attribute classifier.

Our experiments are based on two real-world CES simulators of buildings and wind farm wake. We rigorously evaluate accuracy on held-out systems and show that SysCaps-augmented surrogates have better accuracy than one-hot baselines. We also show generalization beyond the capabilities of traditional regression approaches enabled by the use of text embeddings, e.g., robustness to replacing attributes names with synonyms in test captions. We qualitatively show that text interfaces unlock system design space exploration via language. As there are no standard benchmarks for comparing surrogate modeling performance for CES, we open-source all code and data at `https://github.com/NREL/SysCaps` to facilitate future work.

## 2 RELATED WORK

**Language interfaces for scientific machine learning**: An increasing amount of work is exploring language interfaces for advanced scientific machine learning (SciML) models, including protein representation learning (Xu et al., 2023), protein design (Liu et al., 2023b), and activity prediction for drug discovery (Seidl et al., 2023). LLM-powered natural language interfaces are also being designed for complicated scientific workflows including synchrotron management (Potemkin et al., 2023), automated chemistry labs (Bran et al., 2023), and fluid dynamics workflows (Kumar et al., 2023). We add to this body of work by studying language interfaces for lightweight surrogate models.

**Large language models for regression:** Another line of work asks whether LLMs can perform regression with both text inputs and outputs (numbers encoded as tokens), such as for tabular problems (Dinh et al., 2022) or black-box optimization (Song et al., 2024; Liu et al., 2024). We do not use an LLM to do regression directly, but rather train a lightweight multimodal architecture that

predicts continuous outputs instead of tokens. Moreover, one study (Dinh et al., 2022) found mixed results when comparing to simple gradient-boosted tree baselines and highlighted difficulty with interpolation, raising questions about the effectiveness of LLM-based regression.

**Multimodal text and timeseries forecasting**: Timeseries forecasting aims to predict future values of an input timeseries given past values. Surrogate modeling can be cast as a forecasting problem when the goal is to train a model to emulate a dynamical system and predict its future behavior (e.g., predicting future energy demand from past energy usage). Previous work has explored multimodal timeseries forecasting where auxiliary text data is introduced as covariates to improve the forecasting accuracy (Rodrigues et al., 2019; Emami et al., 2023a; Jin et al., 2024). Notably, Time-LLM (Jin et al., 2024) "reprograms" an LLM to process both text prompts and timeseries. However, our timeseries regression setting (Section 3) differs in that our surrogates are trained to map simulator inputs (e.g., building characteristics and a weather timeseries) to simulator outputs (e.g., energy usage). In our problem, models critically depend on the system information encoded as text, whereas in Time-LLM the text only contains auxiliary information that slightly improves forecasting accuracy. This critical dependence partially motivates our development of a lightweight architecture for fusing text and timeseries embeddings. Also, recent evidence (Merrill et al., 2024; Tan et al., 2024) brings into question whether LLMs are useful at all for reasoning about temporal data.

**Multimodal text and timeseries contrastive pretraining**: Various efforts have explored contrastive pretraining objectives for modeling text and timeseries data (Agostinelli et al., 2023; Huang et al., 2022; Liu et al., 2023a; Zhou et al., 2023). For example, Agostinelli et al. (2023) aligns embeddings of captions that describe the audio. In our setting, captions describe system attributes, and the timeseries inputs are exogenous simulator inputs. Here, the text and timeseries inputs share no information with which to learn a shared embedding space.

**Knowledge-enhanced PDE surrogates**: Numerical simulations of partial differential equations (PDEs) are computationally intensive; thus, much SciML aims to train PDE surrogates which are efficient to evaluate on unseen inputs. Recent work tries to encode knowledge about the PDE into the surrogate to facilitate generalization within and across families of PDEs. Specifically, CAPE (Takamoto et al., 2023) and methods explored in Gupta & Brandstetter (2022) embed equation parameters (i.e., the system attributes) within the architecture to generalize to unseen parameters. Others embed structural knowledge about the PDE into the surrogate model architecture (Rackauckas et al., 2020; Ye et al., 2024) or the loss (Raissi et al., 2019). Concurrent work has explored "PDE captions" (Lorsung et al., 2024; Yang et al., 2024), which are a type of SysCaps for neural PDE surrogates where the system knowledge is PDE equations encoded as text.

## 3 PROBLEM STATEMENT

Our goal is to learn a surrogate $f : \mathcal{X} \times \mathcal{Z} \to \mathcal{Y}$ that regresses the outputs of a simulator $F$ directly from its inputs. We are given a dataset $D$ of pairs of simulator inputs and outputs. The inputs are a deployment scenario $x_{1:T} \in \mathcal{X}$ (a timeseries) and the tabular system attributes $z \in \mathcal{Z}$. The outputs are a timeseries $y_{1:T} \in \mathcal{Y}$. For simplicity, we consider only univariate timeseries outputs in this work ($y_t \in \mathbb{R}$). However, the number of timesteps $T$ may be large ( thousands of steps), the timeseries inputs $\mathcal{X}$ are multivariate, and the mapping $f$ which approximates the simulator is highly nonlinear.

To summarize, we have a timeseries regression problem modeled as $y_{1:T} = f(x_{1:T}, z)$. By conditioning the surrogate on system knowledge $z$, it can potentially generalize to new system configurations. However, learning transferable representations of variable-length, heterogeneous input features such as $z$ is notoriously difficult for deep neural networks, and is a key focus of tabular deep learning (see survey by Badaro et al. (2023)). In our work, we develop and analyze a framework for learning multimodal surrogates where $z$ is encoded as text.

Some simulators may have inputs that are not clearly distinguishable into what is $\mathcal{X}$ and $\mathcal{Z}$, for example, if a dynamical system simulation is configured to be in steady-state or assumes fixed exogenous conditions. In these cases, we allow $\mathcal{X}$ to be a vector of real-valued scalars (a timeseries with $T = 1$), or, simply an empty set (leaving only $\mathcal{Z}$).

**Example:** In many CES, the timeseries $\mathcal{X}$ are exogenous inputs to the system such as weather timeseries consisting of temperature or wind speed. Attributes $\mathcal{Z}$ of a wind farm might include the number of turbines in the wind farm and turbine blade length.

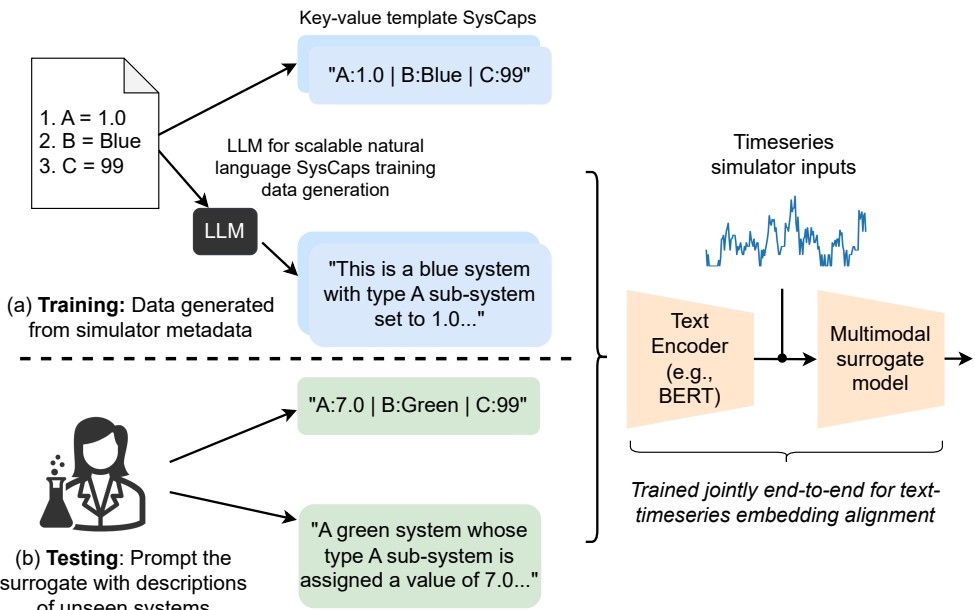

Figure 1: Our pipeline for augmenting multimodal simulation surrogates with language interfaces using "system captions", or SysCaps. SysCaps are text descriptions of knowledge about the system being simulated. In our work, the SysCaps describe the system's characteristics, as found in simulation metadata files. During training (a), we create paired datasets of temporal simulator inputs with key-value template SysCaps or LLM-generated natural language SysCaps. At test time (b), we prompt the surrogate model with one or more key-value template captions or natural language captions. LLMs are only used to generate synthetic training data; we use a lightweight BERT-style text encoder and an efficient long-sequence encoder to keep the computational cost of our surrogate low.

## 4 SYNTHESIZING SYSTEM CAPTIONS (SYSCAPS) WITH LLMS

Our work is motivated by the idea that language interfaces for surrogates represent a path towards improving the accessibility of these models for expert and non-expert users, e.g., when using them for downstream system design tasks (Vaithilingam et al., 2024). Our proposed framework for augmenting surrogates with language interfaces is visualized in Figure 1. During training, we create SysCaps out of system attributes specified in simulation metadata. To create large amounts of synthetic natural language SysCaps, we use LLMs. The ultimate goal is to enable scientists to "chat" with the multi-modal surrogate model at test time via text prompting. In this section, we describe two approaches for converting system attributes into text: key-value templates and natural language.

For the key-value approach, attributes are described as key-value pairs `key:value` and joined by a separator "|" (**SysCaps-kv**). For example, if a simulation has attributes A=1.0 and B=blue, we create the string `A:1.0|B:blue`. Generating these strings is easy to do and incurs a negligible amount of extra computational overhead. In the natural language approach (**SysCaps-nl**, Figure 1), attributes are described in a conversational manner, which we believe is more flexible and expressive than key-value captions and thereby more accessible for non-experts. However, we do not have access to large quantities of natural language descriptions for each system and simulation. We avoid the time-consuming task of enlisting domain experts to create this data by instead prompting a powerful LLM to generate synthetic natural language descriptions given attributes. In our work, we use the open-source LLM `llama-2-7b-chat` (Touvron et al., 2023). The details of the prompt are provided next.

**Prompt design**: We append a carefully written instruction template to a list of system attributes to help guide the LLM in generating a caption via prompting (see Figure 1). The system prompt is: *You are a <CES> expert who provides <CES> descriptions <STYLE>*. The user prompt is: *Write a*

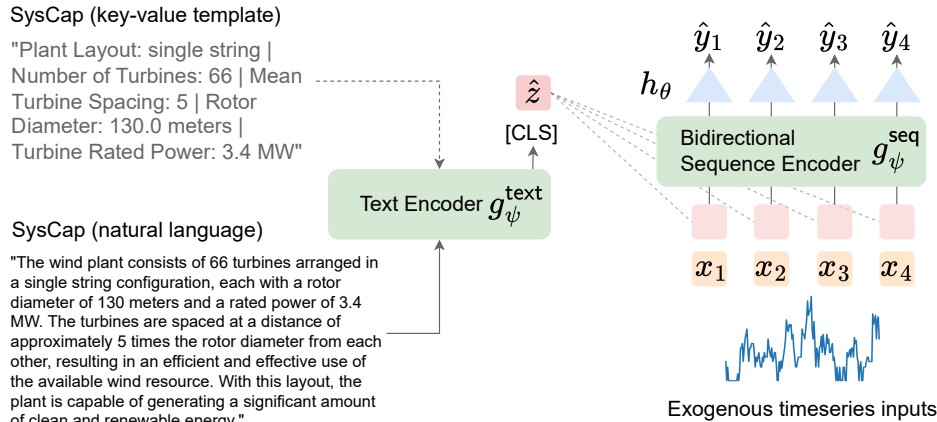

Figure 2: Building blocks of our surrogate model, $f = h_\theta \circ g_\psi$, that includes a multimodal encoder, $g_\psi$, and a top model, $h_\theta$. The multimodal encoder, $g_\psi = g_\psi^{\text{seq}} \circ g_\psi^{\text{text}}$, is a composition of a text encoder, $g_\psi^{\text{text}}$, and a bidirectional sequence encoder, $g_\psi^{\text{seq}}$, for timeseries inputs. The text embedding vector $\hat{z}$ is broadcasted (dashed lines) to create a sequence that is concatenated with the timeseries input. This multimodal sequence is the input to the sequence encoder.

*<CES> description based on the following attributes. Your answer should be <NUM> sentences. Please note that your response should NOT be a list of attributes and should be entirely based on the information provided.* The last part is added to discourage the LLM from changing or omitting attributes. The tags <CES>, <STYLE>, <NUM> are filled in with the CES type (e.g., *buildings*), the style of the description (e.g, *with an objective tone*), and the number of sentences to use in the description (e.g., "4-6"), respectively.

**Attribute subset selection**: Simulations of real-world systems may have attributes that only weakly correlate with the output quantity of interest, or have a large number of attributes, which can be challenging for deep learning approaches. Since the length of a SysCap is proportional to the number of attributes, the computational burden incurred by text-based encodings of attributes can grow significantly in these cases. In these cases, reducing the number of attributes can be handled with classic feature selection methods such as recursive feature elimination (RFE) (Guyon et al., 2002) or by recommendations from domain experts, as a pre-processing step.

## 5  TEXT AND TIMESERIES SURROGATE MODEL

We now describe a lightweight multimodal surrogate model for timeseries regression. The surrogate $f$ (Figure 2) is a composition of a multimodal encoder function $g_\psi : (\mathcal{Z}, \mathcal{X}) \to \{\mathbb{R}^d\}_{1:T}$ and a top model $h_\theta : \mathbb{R}^d \to \mathbb{R}$, where for simplicity, the model parameters $\theta$ are shared across timesteps to predict each timeseries output $y_t$. The training objective is to minimize the expected mean square error averaged over simulation timesteps,

$$\min_{\theta, \psi} \mathbb{E}_{(z, x_{1:T}, y_{1:T}) \sim D} \left[ \frac{1}{T} \sum_{t=1}^{T} \left( [h_\theta(g_\psi(z, x_{1:T}))]_t - y_t \right)^2 \right]. \tag{1}$$

Although more sophisticated loss functions than Eq. 1 could be used to account for predictive uncertainty, we left this extension for future work to simplify our exposition and experiments.

**Multimodal encoder** $g_\psi$: A text encoder $g_\psi^{\text{text}}$ extracts an embedding $\hat{z}$ from a SysCap $z$, then broadcasts and concatenates this embedding with the timeseries inputs to create a sequence of multimodal feature vectors. These features get processed by a bidirectional sequence encoder $g_\psi^{\text{seq}}$ to produce a sequence of time-dependent fused multimodal features $e_{1:T}$, $e_{1:T} = g_\psi^{\text{seq}}(g_\psi^{\text{text}}(z), x_{1:T})$, which are finally used to regress outputs.

**Text encoder** $g_\psi^{\text{text}}$: To encode textual inputs we use pretrained BERT (Devlin et al., 2018) and DistilBERT (Sanh et al., 2019) models that are relatively more efficient than LLMs. We use the model's default pretrained tokenizer. Tokenized sequences are bracketed by `[CLS]` and `[EOS]` tokens, and we use the final activation at the `[CLS]` token position to produce a text embedding $\hat{z} \in \mathbb{R}^d$. Following standard fine-tuning practices, all layers for BERT are fine-tuned while only the last layer of DistilBERT is fine-tuned.

**Bidirectional sequence encoder** $g_\psi^{\text{seq}}$: We broadcast the text embedding $\hat{z}$ to create a sequence of length $T$, $\hat{z} \rightarrow \{\hat{z}_t\}_{t=1}^T$, and concatenate each $\hat{z}_t$ with the timeseries input $x_{1:T}$, $\{\hat{z}_t; x_t\}_1^T$. This simplifies the task of learning timestep-specific correlations between system attributes $\hat{z}$ and timeseries $x_{1:T}$ in the multimodal encoder $g_\psi$. To efficiently embed long timeseries with thousands of timesteps, we explore both bidirectional LSTMs (Hochreiter & Schmidhuber, 1997) and bidirectional SSMs (Goel et al., 2022) for $g_\psi^{\text{seq}}$. Our bidirectional SSM uses stacks of S4 blocks (Gu et al., 2021) without downpooling layers. We use the last layer's hidden states as temporal features $e_{1:T}$ for the top model. If $T = 1$ or for non-sequential surrogate models, we instead use an MLP with residual layers (ResNet MLP) to embed each $\{\hat{z}_t; x_t\}$ per-timestep to get $e_t$.

**Top model** $h_\theta$: The multimodal encoder $g_\psi$ produces $T$ feature vectors $e_{1:T}$. For simplicity, the output $\hat{y}_t$ at each timestep is predicted from $e_t$ by a shared MLP with a single hidden layer.

## 6 EXPERIMENTS

This section presents our experimental results on two real-world CES simulators for buildings (Section 6.1-6.4) and wind farms (Section 6.5). Our experiments study the quality of LLM-generated SysCaps (Section 6.1), accuracy on held-out systems (Section 6.2), generalization under distribution shifts (Section 6.3), show a design space exploration application (Section 6.4), and examine SysCaps prompt augmentation (Section 6.5). All SysCaps are synthetically generated in this work. We provide additional qualitative examples of SysCaps in Appendix B.

**Building stock simulation data:** For the main experiments in Section 6.1-6.4 we train building stock surrogate models for the building energy simulator EnergyPlus (Crawley et al., 2001). Given an annual hourly weather timeseries ($T = 8{,}760$) with 7 variables and a list of tabular building attributes, surrogates predict the building's energy consumption at each hour of the year. Each building initially has 17 attributes which we reduce to 13 with RFE and LightGBM (Ke et al., 2017). We use commercial buildings from the Buildings-900K dataset (Emami et al., 2023b).Commercial building stock surrogates can provide significant speedups compared to EnergyPlus, e.g., 96% (Zhang et al., 2021). Since this dataset only provides energy timeseries, we manually extracted the building configuration and weather timeseries from the End-Use Load Profiles database (Wilson et al.) for each building. Our training set is comprised of 330K buildings, and we use 100 buildings for validation and 6K held-out buildings for testing. We also reserved a held-out set of 10K buildings for RFE. We carefully tune the hyperparameters of all models (details in Appendix A.2).

We created three SysCaps datasets: a "medium" caption length dataset where <NUM> :="4-6", a "short" dataset using 2-3 sentences, and a "long" dataset using 7-9 sentences. The SSMs in our experiments are trained with medium captions. Generating these datasets with `llama-2-7b-chat` used ~1.5K GPU hours on a cluster with 16 NVIDIA A100-40GB GPUs.

### 6.1 EVALUATING CAPTION QUALITY

The LLM that generates natural language SysCaps may erroneously ignore or hallucinate attributes, which may negatively impact downstream performance. To test this, we propose evaluating generated captions by estimating the fraction of attributes which the LLM successfully includes per caption. To compute this metric, we train a multi-class classifier to predict each categorical attribute in a SysCaps from its text embedding. We used a held-out validation set of captions to check that the classifier was not overfitting. The test rate of missing or incorrect attributes is around 9-12% across the "short", "medium", and "long" caption types, with "short" captions having the highest error (Table 2). This increases our confidence that our LLM-based approach for generating natural language SysCaps preserves sufficient information for surrogate modeling.

Table 1: **Accuracy.** We show the mean NRMSE across 3 random seeds. Lower NRMSE is better. Building-hourly is the NRMSE normalized per building and per hour. Stock-annual first sums the predictions and targets over all buildings and hours, before computing the NRMSE (equivalent to the normalized mean bias error—see Appendix A.1). When using a BERT encoder instead of DistilBERT, SysCaps surrogates achieve better accuracy than one-hot baselines. We trained one SSM without any attribute information (Attribute Encoding "X") as an ablation study; the poor accuracy shows that our multimodal architecture successfully learns to fuse the text-based attribute and timeseries inputs.

| Model | Text Encoder | Attribute Encoding | Buildings-Hourly (NRMSE) | Stock-Annual (NRMSE) |
|---|---|---|---|---|
| SSM | - | X | $1.712_{\pm 0.003}$ | $0.658_{\pm 0.008}$ |
| SSM | - | onehot | $0.450_{\pm 0.019}$ | $0.041_{\pm 0.021}$ |
| LSTM | - | onehot | $0.449_{\pm 0.025}$ | $0.045_{\pm 0.024}$ |
| ResNet | - | onehot | $0.634_{\pm 0.009}$ | $0.072_{\pm 0.008}$ |
| LightGBM | - | onehot | $0.679_{\pm 0.014}$ | $0.094_{\pm 0.003}$ |
| SSM | DistilBERT | SysCaps-nl | $0.532_{\pm 0.010}$ | $0.069_{\pm 0.010}$ |
| SSM | BERT | SysCaps-nl | $0.543_{\pm 0.011}$ | $0.035_{\pm 0.005}$ |
| SSM | DistilBERT | SysCaps-kv | $0.454_{\pm 0.012}$ | $0.046_{\pm 0.003}$ |
| SSM | BERT | SysCaps-kv | $0.450_{\pm 0.007}$ | $\mathbf{0.020}_{\pm 0.012}$ |
| LSTM | DistilBERT | SysCaps-kv | $0.489_{\pm 0.021}$ | $0.063_{\pm 0.005}$ |
| LSTM | BERT | SysCaps-kv | $\mathbf{0.439}_{\pm 0.037}$ | $0.022_{\pm 0.011}$ |
| ResNet | DistilBERT | SysCaps-kv | $0.633_{\pm 0.020}$ | $0.081_{\pm 0.011}$ |
| ResNet | BERT | SysCaps-kv | $0.670_{\pm 0.043}$ | $0.049_{\pm 0.015}$ |

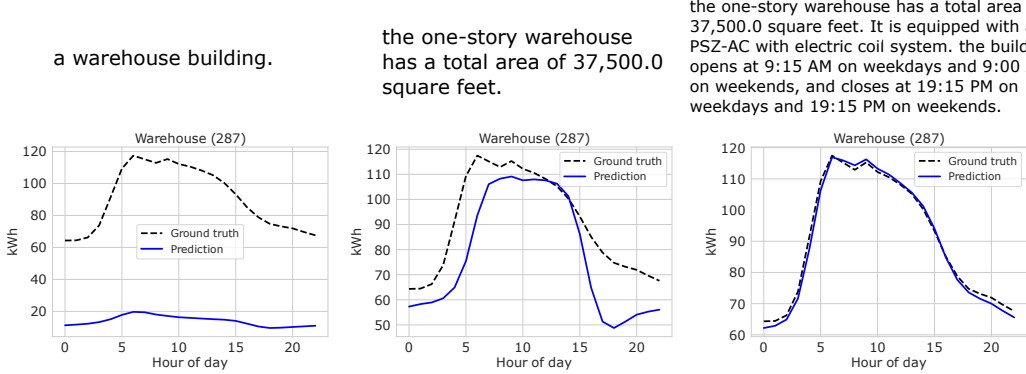

Figure 3: **System captions unlock text-prompt-style surrogate modeling for complex systems.** We show building stock daily load profiles aggregated for Warehouse building type, created with caption templates. From left to right, we use captions with one, three, and six attributes.

## 6.2 ACCURACY ON HELD-OUT SYSTEMS

In this section, we compare the accuracy of SysCaps surrogate models that vary by textual attribute encoding (key-value (-kv) and natural language (-nl)), text encoder (BERT, DistilBERT), and sequence encoder (LSTM, SSM). We also train SysCaps surrogates with a non-sequential encoder (ResNet), without attribute inputs, and without text encoder fine-tuning, as ablations. Baselines are a tuned LightGBM Gradient Boosting Decision Tree and a ResNet, LSTM, and SSM surrogate with onehot encoded attributes. Following Emami et al. (2023b), we use the normalized root mean square error (NRMSE) metric averaged across 3 random seeds.

**Does the sequential architecture matter?** *Yes*—Table 1 shows that the LSTM and SSM encoders outperform both the ResNet and our carefully tuned LightGBM baseline, and the SSM outperforms the LSTM. **How do different system attribute encoding approaches compare?** Surprisingly,

Table 2: **Caption quality**. We estimate the presence of each attribute in a SysCap, measured by the average test accuracy of a multi-class classifier trained to predict each categorical attribute. Our metric suggests ∼9-12% of attributes are missing or incorrect per SysCap, due to errors made by `llama-2-7b-chat`.

| Caption length (13 attributes) | Accuracy (%) |
|---|---|
| Short (2-3 sent.) | 88.90 |
| Medium (4-6 sent.) | 90.90 |
| Long (7-9 sent.) | 90.38 |

Table 3: **SysCaps zero-shot length generalization**. NRMSE is per-building-hourly. Results are for the SSM model trained with medium-length SysCaps and evaluated zero-shot on short and long captions.

| SysCaps length | NRMSE |
|---|---|
| Short | $0.57_{\pm 0.02}$ |
| Medium | $\mathbf{0.53}_{\pm 0.01}$ |
| Long | $0.64_{\pm 0.02}$ |

Table 4: **Generalization to attribute synonyms.** We quantify how text embeddings make our models robust to the use of attribute synonyms. The metric is the difference in NRMSE between the original caption and the modified caption. We bold the best method, i.e, closest to 0. Column 3 replaces the building type with a synonym, column 4 removes the building type and sub-type attributes from the caption, and column 5 randomly swaps the correct building type attributes with incorrect ones.

| Building Type | Synonym | With Synonym | Without Building Type | Random Building Type |
|---|---|---|---|---|
| FullServiceRestaurant | FineDiningRestaurant | $\mathbf{0.52}_{\pm 0.05}$ | $0.93_{\pm 0.01}$ | $1.17_{\pm 0.07}$ |
| RetailStripmall | ShoppingCenter | $\mathbf{0.01}_{\pm 0.00}$ | $0.68_{\pm 0.02}$ | $0.28_{\pm 0.04}$ |
| Warehouse | StorageFacility | $\mathbf{0.35}_{\pm 0.30}$ | $0.55_{\pm 0.31}$ | $4.02_{\pm 0.32}$ |
| RetailStandalone | ConvenienceStore | $\mathbf{0.00}_{\pm 0.01}$ | $0.30_{\pm 0.04}$ | $0.40_{\pm 0.03}$ |
| SmallOffice | Co-WorkingSpace | $0.03_{\pm 0.01}$ | $\mathbf{0.02}_{\pm 0.02}$ | $1.95_{\pm 0.30}$ |
| PrimarySchool | ElementarySchool | $\mathbf{0.00}_{\pm 0.01}$ | $0.38_{\pm 0.02}$ | $0.52_{\pm 0.17}$ |
| MediumOffice | Workplace | $0.08_{\pm 0.02}$ | $\mathbf{0.03}_{\pm 0.04}$ | $0.91_{\pm 0.11}$ |
| SecondarySchool | HighSchool | $\mathbf{-0.01}_{\pm 0.04}$ | $0.52_{\pm 0.06}$ | $0.67_{\pm 0.33}$ |
| Outpatient | MedicalClinic | $\mathbf{0.02}_{\pm 0.01}$ | $0.55_{\pm 0.09}$ | $0.32_{\pm 0.06}$ |
| QuickServiceRestaurant | FastFoodRestaurant | $\mathbf{0.10}_{\pm 0.07}$ | $0.83_{\pm 0.01}$ | $0.85_{\pm 0.01}$ |
| LargeOffice | OfficeTower | $\mathbf{0.12}_{\pm 0.13}$ | $0.23_{\pm 0.03}$ | $0.29_{\pm 0.11}$ |
| LargeHotel | Five-Star Hotel | $\mathbf{0.03}_{\pm 0.01}$ | $0.46_{\pm 0.06}$ | $0.29_{\pm 0.09}$ |
| SmallHotel | Motel | $\mathbf{0.26}_{\pm 0.07}$ | $0.88_{\pm 0.07}$ | $0.70_{\pm 0.14}$ |
| Hospital | HealthcareFacility | $\mathbf{0.03}_{\pm 0.04}$ | $0.62_{\pm 0.12}$ | $0.20_{\pm 0.07}$ |

SysCaps-kv achieves the best accuracy overall at the stock-annual aggregation level (equivalent to the normalized mean bias error—see Appendix A.1) and comparable building-hourly accuracy to one-hot baselines. The SysCaps-nl models have slightly worse accuracy than SysCaps-kv, yet they comfortably outperform the non-sequential models (including LightGBM) and the one-hot baselines at the stock-annual aggregation level. We initially expected to see a non-negligible drop in regression accuracy for SysCaps models, even key-value SysCaps, because the text encoder has to compress the caption into a single embedding vector, which causes information loss. However, the BERT encoder is expressive enough to mitigate this. We observe that stock-annual NRMSE reduces by about half when switching from DistilBERT to BERT. We believe the gap between key-value and natural language is mostly explained by caption quality (Table 2). We ablate the importance of using attributes by training an SSM baseline without attributes (Attribute Encoding "X"). This model is unable to learn this task. We trained a SysCaps-nl model without fine-tuning BERT, but it does poorly (stock-annual NRMSE of 0.356), showing the importance of fine-tuning for multi-modal alignment.

Figure 3 qualitatively shows how a SysCaps model performs with natural language captions (created with a sentence template) provided to the model. Note that we did not train our models on any captions with missing attributes. Prediction accuracy improves as more information is given; notably, there is a large jump in accuracy once the building square footage is known.

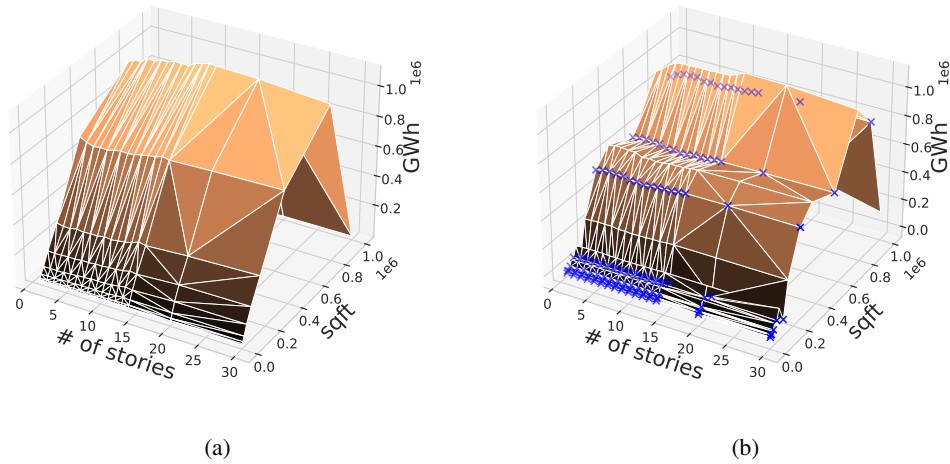

(a)                                 (b)

Figure 4: **Design space exploration using language**. a) We show that the model has learned a physically plausible relationship between building square footage (sqft) and number (#) of stories. b) Failure case: When tested on unseen values of sqft (blue crosses), the model's predictions appear to be physically implausible—the model underestimates the energy consumption at these sqft values.

## 6.3 CAPTION GENERALIZATION

**Length generalization:** We assess how accuracy varies when surrogates are provided with natural language SysCaps having different lengths than seen during training. We evaluate zero-shot generalization to the short and long captions. The results (Table 3) show a small increase in error for shorter captions with a larger increase in error for longer captions. However, the error on long captions remains lower than the error achieved by our tuned LightGBM baseline.

**Attribute synonyms:** To quantitatively evaluate the extent to which natural language SysCaps surrogates gain a level of robustness to distribution shifts such as word order changes, synonyms, or writing style (Hendrycks et al., 2020), we created captions for the held-out systems where the "building type" attribute is replaced by a synonym. We avoid biasing the choice of building type synonym by 5-shot prompting `llama-2` to suggest the synonyms. There are two baselines we compare the synonym caption accuracy against: 1) accuracy when testing the model on captions with the building type attribute removed, and 2) accuracy when testing the model on captions with a random building type. Examples and results are shown in Table 4, where for 11/14 building type synonyms the increase in NRMSE is less than 13%, while the average increase for the two baselines is 54% and 90%, respectively.

## 6.4 DESIGN SPACE EXPLORATION APPLICATION USING LANGUAGE

We visualize in Figure 4a a use of SysCaps to conduct a sensitivity analysis on two system attributes, as might be performed for an early-stage design space exploration task. We use a simple template to create a caption for each test building that enumerates all combinations of the number of stories and square footage attributes, totaling 160 configurations; the entire analysis requires simulating 960K buildings, and took 1 hour on a single NVIDIA A100 GPU. We observe that the model has indeed learned physically plausible relationships between these two attributes. However, the model fails to predict the energy usage for buildings over 100K square feet—such buildings are in the "long tail" of the training data distribution. Figure 4b also shows a failure case where the model underestimates energy usage at unseen numeric building square footage attribute values.

## 6.5 PROMPT AUGMENTATION: WIND FARM WAKE

This experiment uses the Wind Farm Wake Modeling Dataset (Ramos et al., 2023), made with the FLORIS simulator, to train a surrogate to predict a wind farm's power generation in steady-state

atmospheric conditions. The speed-up provided by surrogate models for downstream optimization use-cases is $\sim700\times$ (Harrison-Atlas et al., 2024). The difficulty of this task is in modeling losses due to wake effects, given only a coarse description of the wind farm layout. There are three numeric simulator inputs $x$ specifying atmospheric conditions, and five system attributes which include categorical variables indicating wind farm shape (four different layout types), number of turbines, and average turbine spacing. We do not use RFE. In this dataset, there are only 500 unique system configurations (split 3:1:1 for train, val, test), although each configuration is simulated under 500 distinct atmospheric conditions.

We explore generating multiple captions for each system configuration through prompt augmentation to increase diversity. Specifically, we replace the <STYLE> tag in the prompt with phrases encouraging different description styles, e.g., *with an objective tone*, *with an objective tone (creative paraphrasing is acceptable)*, *to a colleague*, and *to a classroom*. The simulation is run assuming steady-state conditions (i.e., time-independent), so we tune hyperparameters for and train the non-sequential ResNet models. The ResNet baseline with one-hot encoded attributes suffers from severe overfitting (Table 5), likely due to the small number (300) of training systems, whereas the SysCaps models generalize better to unseen systems. This suggests SysCaps can have a regularizing effect in small data settings. Notably, the prompt augmentation helps the natural language SysCaps model to achieve the lowest NRMSE.

Table 5: **Wind farm surrogate accuracy.** The base architecture is ResNet. Average across 3 random seeds. SysCaps-nl* does not use prompt augmentation.

| Model | NRMSE |
|---|---|
| LightGBM | $0.196_{\pm 0.000}$ |
| one-hot | $0.212_{\pm 0.009}$ |
| SysCaps-kv | $0.054_{\pm 0.024}$ |
| SysCaps-nl* | $0.038_{\pm 0.001}$ |
| SysCaps-nl | $\mathbf{0.035}_{\pm 0.001}$ |

## 7 CONCLUSION

In this work, we introduced a lightweight, multimodal text and timeseries surrogate models for complex energy systems such as buildings and wind farms, and described a process for using LLMs to synthesize natural language descriptions of such systems, which we call SysCaps. Our experiments showcase SysCaps-augmented surrogates that achieve better accuracy than standard feature engineering (e.g., one-hot encoding) while also enjoying the advantages of using text embeddings such as robustness to caption paraphrasing (e.g., synonyms of attributes). For a problem with only a *small* number of training systems available, we showed that SysCaps-nl prompt augmentation has a regularizing effect that helps mitigate overfitting. Overall, these results underscore that language is a viable interface for interacting with real-world surrogate models.

**Limitations**: Current BERT-style tokenizers struggle with numerical values (Wallace et al., 2019); for one example, they interpolate poorly to unseen numbers (Figure 4b). For another, because `llama-2-7b-chat` tends to add a comma to large numbers (e.g., $200,000$) when generating SysCaps, we found that our surrogates failed to understand large numbers without commas (had high error). Orthogonal research on improving number encodings for language model inputs (Golkar et al., 2023; Yan et al., 2024) can benefit our framework. Another potential concern is with creating SysCaps for simulators with a large number of attributes (e.g., over 100). A more powerful LLM than `llama-2-7b-chat` with a longer context window may be needed in this case. In general, we expect that more powerful LLMs will further improve the quality of the training captions.

**Future work**: A future extension of this work might explore how to use language to also interface with the timeseries simulator inputs, possibly through summary statistics. For example, to study how the complex system behaves when the average exogenous temperature is increased by five degrees. It is natural to expect that non-experts may benefit more from our approach if the LLM is also instructed to simplify the simulator metadata or to provide explanations of technical concepts. Conducting interactive evaluations with non-experts will be important to obtain feedback for further improving the approach. Likewise, conducting a series of studies with domain scientists to evaluate the quality of SysCaps in the context of, e.g., system design optimization is promising. We did not conduct user studies in this work, as we first aimed to establish technical feasibility of this surrogate modeling approach. Finally, an important question is how we might create *surrogate foundation models* that generalize not only across system configurations for a single simulator, but also generalize across different simulators.

## ACKNOWLEDGMENTS

This work was authored by the National Renewable Energy Laboratory (NREL), operated by Alliance for Sustainable Energy, LLC, for the U.S. Department of Energy (DOE) under Contract No. DE-AC36-08GO28308. This work was supported by the Laboratory Directed Research and Development (LDRD) Program at NREL. The views expressed in the article do not necessarily represent the views of the DOE or the U.S. Government. The U.S. Government retains and the publisher, by accepting the article for publication, acknowledges that the U.S. Government retains a nonexclusive, paid-up, irrevocable, worldwide license to publish or reproduce the published form of this work, or allow others to do so, for U.S. Government purposes. The research was performed using computational resources sponsored by the Department of Energy's Office of Energy Efficiency and Renewable Energy and located at the National Renewable Energy Laboratory.

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

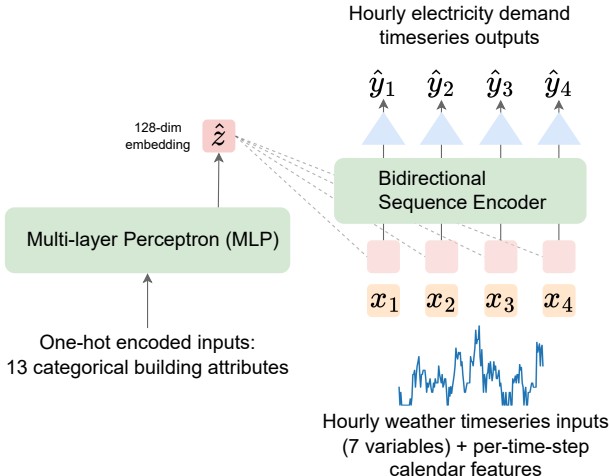

Figure 5: A visualization of the sequential surrogate model baseline with one-hot encoded system attributes for the buildings experiment.

## A ADDITIONAL EXPERIMENT DETAILS

### A.1 METRICS

We use the normalized root mean square error (NRMSE) to capture the accuracy of the surrogate model. NRMSE is also known as (CV)RMSE.

The **building-hour NRMSE** is where the NRMSE is normalized by building and by hour, where $B$ is the number of buildings in the building stock and $T$ is the number of hours in a year:

$$:= \frac{1}{\frac{1}{BT}\sum y_t^b} \sqrt{\frac{1}{BT}\sum_{b=1,t=1}^{B,T}(y_t^b - \hat{y}_t^b)^2}. \tag{2}$$

The **stock-annual NRMSE** is where the NRMSE is normalized by the annual stock energy consumption:

$$:= \frac{1}{\sum y_t^b} \sqrt{\left(\left(\sum_{b=1,t=1}^{B,T} y_t^b\right) - \left(\sum_{b=1,t=1}^{B,T} \hat{y}^b\right)\right)^2}. \tag{3}$$

Notice that the square and square root cancel, making the stock-annual NRMSE equivalent to the normalized mean bias error.

We use the AdamW (Loshchilov & Hutter, 2017) optimizer with $\beta_1 = 0.9$, $\beta_2 = 0.98$, $\epsilon = 1\text{e-}9$, and weight decay of 0.01 for all experiments. The early stopping patience is 50 for all experiments. All models are trained with a single NVIDIA A100-40GB GPU. The longest training runs take 1-2 days and the shortest 2-3 hours.

### A.2 HYPERPARAMETERS

See Table 6 for hyperparameter sweep details for the buildings experiments and Table 7 for hyperparameter sweep details for the wind farm experiments.

#### A.2.1 BUILDINGS

There are 13 attributes after RFE, which are one-hot encoded into a 336-dimensional feature vector that gets embedded into 128 dimensions, whereas the text embeddings are 768-dimensional. We

Table 6: Building stock surrogate model hyperparameters.

| Model | Hyperparameter | Grid search space | Best values |
|---|---|---|---|
| LightGBM | Learning rate | From 0.01 to 0.1 | 0.066 |
| | Number of leaves | From 40 to 150 | 149 |
| | Subsample | From 0.05 to 1.0 | 0.178 |
| | Feature fraction | From 0.05 to 1.0 | 0.860 |
| | Min number of data in one leaf | From 1 to 100 | 12 |
| ResNet (one-hot) | Hidden layers size | 256, 1024, 2048 | 1024 |
| | Number of layers | 2, 8 | 2 |
| | Batch size | 128, 256, 512 | 512 |
| | Learning rate | 0.0001, 0.0003, 0.001 | 0.0003 |
| ResNet (SysCaps) | Hidden layers size | 256, 1024, 2048 | 256 |
| | Number of layers | 2, 8 | 8 |
| | Batch size | 128, 256, 512 | 256 |
| | Learning rate | 0.0001, 0.0003, 0.001 | 0.0003 |
| Bidirectional LSTM (one-hot) | [Hidden layer size, Batch size] | [128, 64], [512, 32], [1024, 32] | [128, 64] |
| | Number of layers | 1, 3, 4, 6, 8 | 4 |
| | MLP dimension | 256 | 256 |
| | Learning rate | 0.00001, 0.0003, 0.001 | 0.001 |
| Bidirectional LSTM (SysCaps) | [Hidden layer size, Batch size] | [128, 64], [512, 32], [1024, 32] | [512, 32] |
| | Number of layers | 1, 3, 4, 6, 8 | 1 |
| | MLP dimension | 256 | 256 |
| | Learning rate | 0.00001, 0.0003, 0.001 | 0.0003 |
| Bidirectional S4 (one-hot) | [Hidden layer size, Num. layers] | [64,8] , [128,4] | [128,4] |
| | MLP dimension | 256 | 256 |
| | Batch size | 32, 64 | 64 |
| | Learning rate | 1e-5, 3e-4, 1e-3 | 3e-4 |
| Bidirectional S4 (SysCaps) | [Hidden layer size, Num. layers] | [64,8] , [128,4] | [128, 4] |
| | MLP dimension | 256 | 256 |
| | Batch size | 32, 64 | 32 |
| | Learning rate | 1e-5, 3e-4, 1e-3 | 3e-4 |

Table 7: Wind farm wake surrogate model hyperparameters.

| Model | Hyperparameter | Grid search space | Best values |
|---|---|---|---|
| LightGBM | Learning rate | From 0.01 to 0.1 | 0.039 |
| | Number of leaves | From 40 to 120 | 108 |
| | Subsample | From 0.6 to 1.0 | 0.963 |
| | Feature fraction | From 0.6 to 1.0 | 0.997 |
| | Min number of data in one leaf | From 20 to 100 | 96 |
| ResNet (one-hot) | Hidden layers size | [256,1024] | 256 |
| | Number of layers | [2,8] | 2 |
| | Batch size | [128,256] | 128 |
| | Learning rate | [1e-5, 3e-4, 1e-3] | 1e-5 |
| ResNet (SysCaps-kv) | Hidden layers size | [256,1024] | 1024 |
| | Number of layers | [2,8] | 8 |
| | Batch size | [128,256] | 256 |
| | Learning rate | [1e-5,3e-4,1e-3] | 3e-4 |
| ResNet (SysCaps-nl) | Hidden layers size | [256,1024] | 1024 |
| | Number of layers | [2,8] | 8 |
| | Batch size | [128,256] | 256 |
| | Learning rate | [1e-5,3e-4,1e-3] | 1e-5 |

concatenate cyclically encoded calendar features to 7 weather variables, creating a 103-dimensional vector (Emami et al., 2023b). For the one-hot models (Figure 5), this creates a 128 + 103 = 231 dimensional input for the bidirectional sequence encoder, and for the text models, it is a 768 + 103 = 871 dimensional input.

---

**Complex Energy System: Commercial Building**

---

**Instruction:** <CES>, <STYLE>, <NUM>

[System] You are a building expert who provides building descriptions with an objective tone.
[User] Write a building description based on the following attributes. Your answer *should* be 4-6 sentences. Please note that your response should NOT be a list of attributes and should be entirely based on the information provided.

**System Knowledge:** Tabular attributes

building_type: RetailStandalone, number_of_stories: 1.0, sqft: 7500.0, hvac_system_type: PSZ-AC with gas coil, weekday_opening_time: 12:15 PM, weekend_opening_time: 10:30 AM, unoccupied cooling temperature set point difference from occupied state: 0.0, occupied cooling temperature set point: 72.0, unoccupied heating temperature set point difference from occupied state: 0.0, occupied heating temperature set point: 67.0, weekday_closing_time: 21:30 PM, weekend_closing_time: 16:15 PM

---

**SysCap:** Natural language

The retail standalone building is a single-story structure with a total area of approximately 7,500 square feet. The building is equipped with a PSZ-AC system with a gas coil, which provides heating and cooling for the space. The building is open from 12:15 PM on weekdays and 10:30 AM on weekends, with closing times of 21:30 PM on weekdays and 16:15 PM on weekends. The temperature set points for both heating and cooling are set at 67°F and 72°F, respectively, when the space is occupied. However, there is no difference in the temperature set points when the space is unoccupied.

---

Figure 6: Natural language building SysCap. The instruction and list of key-value attributes (first two paragraphs) are provided to the LLM as the prompt. We observe that the LLM automatically converts numbers to more human-interpretable descriptions (e.g., the number of stories is changed from 1.0 in the prompt to "single-story"). The LLM also succinctly (and correctly) states that "there is no difference in the temperature set points when the space is unoccupied" in the output by summarizing the "unoccupied heating/cooling temperature set point difference from occupied state: 0.0" attributes.

**LightGBM:** As LightGBM does not support batch training out of the box, the entire training data needs to be loaded into the memory to train a LightGBM model. With the train dataset containing 340k buildings, each with 8759 hours and 347 features, we randomly extract 438 hours per building (which is about 5% of total hours) to limit memory usage. This results in $340,000 \times 438$ hours in total for the train dataset, which consumes about 380 GB of memory when being loaded into a NumPy object. For the validation and test splits, we retain the full number of hours per building. The LightGBM model is tuned with Optuna (Akiba et al., 2019) across 30 trials and achieves the best validation NRMSE of 0.667.

**Multi-class attribute classifier:** We implement the classifier on top of the text encoder by adding a linear layer for each attribute type, where this layer predicts logits for each attribute's classes. We use AdamW with a learning rate of 3e-4, early stopping with patience 5, batch size 128, and max epochs 100. We do not freeze the text encoder weights.

### A.2.2  WIND FARM

**LightGBM:** The training, validation, and test split for the wind dataset gives us datasets of size 148,650, 49,600, and 49,250 respectively with 190 features after one-hot encoding. The training dataset is loaded into memory to train the LightGBM model. We use Optuna (Akiba et al., 2019) to tune the hyperparameters and the best validation NMRSE achieved is 0.189.

## B  SYSCAPS PROMPTS

In this section we visualize examples of prompts and the corresponding SysCaps LLM outputs for the buildings simulator (Figures 6, 7, and 8) and the wind farm simulator (Figures 9, 10, and 11).

Figure 7: Natural language building SysCap. The LLM makes a subtle logical error here—the unoccupied heating temperature set point difference from occupied state suggests the set point should be *lower* by 10 degrees, not higher (highlighted in blue). Logical errors such as this may have contributed to the slightly worse accuracy of natural language SysCaps compared to key-value SysCaps. Collaborating with domain experts able to validate the LLM's outputs is important for catching these errors. We expect that using more powerful LLMs will help reduce logical errors.

## C QUALITATIVE EXAMPLES OF TEST PREDICTIONS

We visualize in Figure 12 the predicted energy timeseries for one test building by the SysCaps-nl and SysCaps-kv models alongside the weather timeseries and each SysCaps type (key-value, short, medium, and long). We also show test predictions for a wind farm (Figure 13) alongside the key-value SysCaps and each style-augmented natural language SysCaps.

---

**Complex Energy System: Commercial Building**

---

**Instruction**: <CES>, <STYLE>, <NUM>

[System] You are a building expert who provides building descriptions with an objective tone.
[User] Write a building description based on the following attributes. Your answer *should* be 4-6 sentences. Please note that your response should NOT be a list of attributes and should be entirely based on the information provided.

**System Knowledge:** Tabular attributes

building_subtype: strip_mall_restaurant30, building_type: RetailStripmall, number_of_stories: 2.0, sqft: 17500.0, hvac_system_type: PSZ-AC with gas coil, weekday_opening_time: 8:15 AM, weekend_opening_time: 11:00 AM, unoccupied cooling temperature set point difference from occupied state: 6.0, occupied cooling temperature set point: 72.0, unoccupied heating temperature set point difference from occupied state: 7.0, occupied heating temperature set point: 68.0, weekday_closing_time: 17:45 PM, weekend_closing_time: 22:30 PM

---

**SysCap:** Natural language

The strip mall restaurant is a two-story building with a total floor area of 17,500 square feet. The building is equipped with a PSZ-AC system with a gas coil, which provides heating and cooling. The building is open from 8:15 AM to 11:00 AM on weekdays and 11:00 AM to 22:30 PM on weekends. During unoccupied periods, the cooling temperature set point is 6.0 degrees lower than the occupied state, while the heating temperature set point is 7.0 degrees lower. The building is well-insulated and energy-efficient, with a weekday opening time of 8:15 AM and a weekend opening time of 11:00 AM.

Figure 8: Natural language building SysCap. The LLM confuses the weekday closing time with the weekend open time (highlighted in blue).

---

**Complex Energy System: Wind Farm**

---

**Instruction**: <CES>, <STYLE>, <NUM>

[System] You are a wind energy expert describing a wind plant to a colleague.
[User] Write a wind plant description based on the following attributes. Your answer *should* be 4-6 sentences. Please note that your response should NOT be a list of attributes and should be entirely based on the information provided.

**System Knowledge:** Tabular attributes

Plant Layout: cluster, Number of Turbines: 123, Mean Turbine Spacing: 5, Rotor Diameter: 130.0 meters, Turbine Rated Power: 3.4 MW

---

**SysCap:** Natural language

The wind plant is laid out in a cluster configuration, with 123 turbines positioned at a spacing of approximately 5 times the rotor diameter. Each turbine features a rotor diameter of 130 meters and is rated to produce 3.4 megawatts of power. The plant's layout and turbine design work together to optimize energy production and minimize visual impact.

Figure 9: Natural language wind farm SysCap.

Figure 10: Natural language wind farm SysCap.

Figure 11: Natural language wind farm SysCap with a logical error where it says "total installed capacity", but it just states the capacity of a single turbine (highlighted in blue). The correct total installed capacity is number of turbines times 3.4, or 431.8 MW.

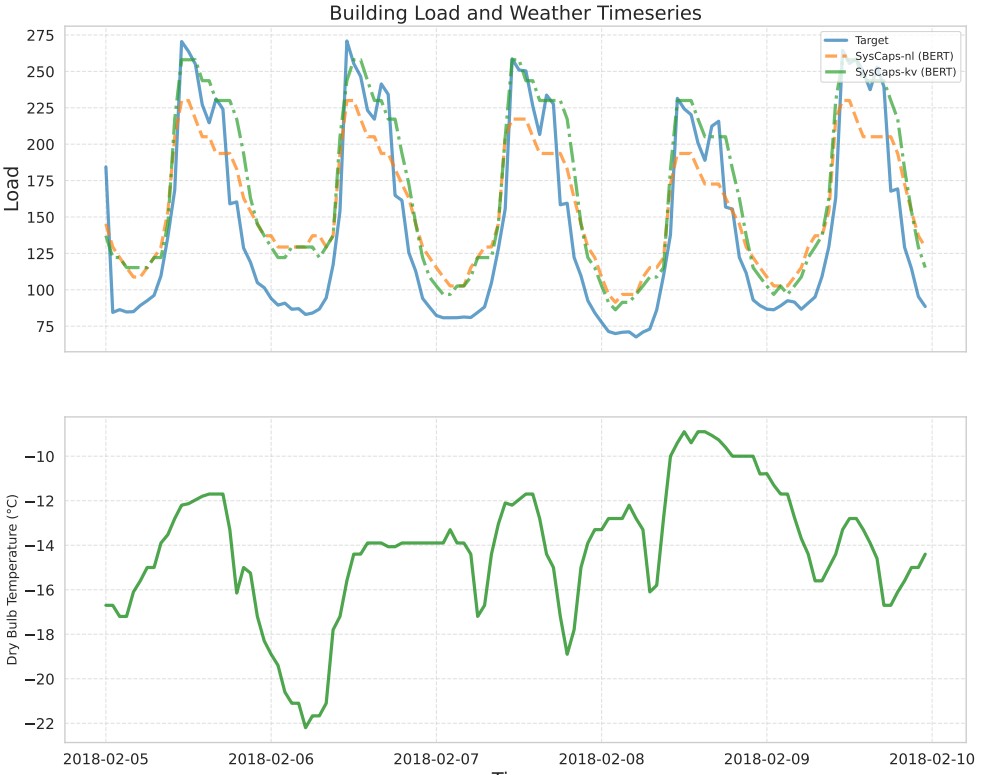

**Medium**: The strip mall restaurant is a single-story retail building with a total floor area of 17,500 square feet. The building is equipped with a PSZ-AC system with a gas coil, which provides heating and cooling to the space. The building is open from 9:15 AM to 10:45 AM on weekdays and 10:45 AM to 19:15 PM on weekends. During unoccupied periods, the cooling temperature set point is set at 73 degrees Fahrenheit, while the heating temperature set point is set at 68 degrees Fahrenheit. The building has a moderate temperature difference between occupied and unoccupied states, with a cooling temperature difference of 3 degrees Fahrenheit and a heating temperature difference of 0 degrees Fahrenheit.

**Short :** The strip mall restaurant is a single-story building with a total area of 17,500 square feet. The HVAC system is a PSZ-AC with gas coil, and the temperature set points for both heating and cooling are set at 68°F and 73°F, respectively, during unoccupied periods. The building opens at 9:15 AM on weekdays and 10:45 AM on weekends, and closes at 5:30 PM on weekdays and 7:15 PM on weekends.

**Long :** The strip mall restaurant is a single-story retail building with a total floor area of approximately 17,500 square feet. The building is equipped with a PSZ-AC with gas coil HVAC system, which provides heating and cooling to the space. During weekdays, the restaurant opens at 9:15 AM and closes at 5:30 PM, while on weekends, it opens at 10:45 AM and closes at 6:00 PM. The building is unoccupied during the night, from 17:30 PM to 10:45 AM on weekdays and from 19:15 PM on weekends. The cooling temperature set point during occupancy is set at 73 degrees Fahrenheit, while the heating temperature set point is set at 68 degrees Fahrenheit. The building has a moderate temperature difference between occupancy and unoccupancy, with a difference of 3 degrees Fahrenheit in cooling and 0 degrees Fahrenheit in heating.

**Keyvalue :** building_subtype:strip_mall_restaurant20|building_type:RetailStripmall|number_of_stories:1.0|sqft:17500.0|hvac_system_type:PSZ-AC with gas coil|weekday_operating_hours:8.25|weekday_opening_time:9.25|weekend_operating_hours:8.5|weekend_opening_time:10.75|tstat_clg_delta_f:3.0|tstat_clg_sp_f:73.0|tstat_htg_delta_f:0.0|tstat_htg_sp_f:68.0

Figure 12: (Top) Predicted hourly energy consumption for a randomly selected week. (Second from top) We visualize 1 out of 7 input weather timeseries, Dry Bulb Temperature. The text boxes show the medium, short, long, and key-value SysCaps created for this particular test building.

Layout214, True: 318.93 MW, SysCaps-nl: 319.50 MW, SysCaps-kv: 315.75 MW

**< STYLE >** : with an objective tone. Creative paraphrasing is acceptable
The wind plant features a layout of multiple strings, with 94 turbines positioned across the site. Each turbine stands at a height of 130 meters, with a rotor diameter that is four times that size. The turbines are spaced at a distance of approximately 400 meters from each other, allowing for optimal wind flow and energy production. With a rated power of 3.4 megawatts, each turbine is capable of generating a significant amount of electricity, making the plant a significant contributor to the local energy grid.

**< STYLE >** : with an objective tone
The wind plant features a layout of multiple strings, with 94 turbines installed across the site. Each turbine has a rotor diameter of 130 meters, and the turbines are spaced at an average distance of four times the rotor diameter apart. The turbines are rated to produce 3.4 megawatts of power, making this a significant contributor to the local renewable energy mix.

**< STYLE >** : to a colleague
The wind plant consists of multiple strings of turbines, with 94 turbines in total. Each turbine has a rotor diameter of 130 meters and is spaced approximately 4 times that distance apart, resulting in a total installed capacity of 3.4 megawatts. The plant takes advantage of the consistent wind patterns in the area to generate clean and renewable energy.

**< STYLE >** : to a classroom
The wind plant consists of multiple strings of turbines stretched across the landscape, with a total of 94 turbines standing tall and proud. Each turbine has a rotor diameter of 130 meters, giving it a massive presence on the horizon. The turbines are spaced at a distance of approximately four times the rotor diameter from each other, creating a uniform and efficient layout. With each turbine capable of generating 3.4 megawatts of power, the plant has a total capacity of over 300 megawatts, enough to power thousands of homes and businesses. As the wind blows, the turbines spin effortlessly, harnessing the natural energy of the wind to generate clean and sustainable electricity.

keyvalue
Plant Layout:multiple strings|Number of Turbines:94|Mean Turbine Spacing:4|Rotor Diameter:130.0 meters|Turbine Rated Power:3.4 MW

Figure 13: We show the LLM-generated SysCaps using four different styles as a prompt augmentation strategy for test wind farm layout # 214. The key-value SysCaps is shown at the bottom. Model predictions are shown at the top next to the true ground truth value.

