# OpenReview forum: "SysCaps: Language Interfaces for Simulation Surrogates of Complex Systems"
_ICLR.cc/2025/Conference — ICLR 2025 Poster_

### Official Review · Reviewer_Gxzq · 2024-10-31

**Soundness:** 3
**Presentation:** 3
**Contribution:** 2
**Rating:** 8
**Confidence:** 2

**Summary:**

This paper studies the use of natural language captions as inputs to surrogate models that simulate "complex energy systems". These natural language captions describe the features of the system being simulated. The task is to predict a timeseries of some variable of interest that depends on these features and some other independent variable that is fed as a time series. The paper introduces an architecture that fuses the textual description with the time series data to achieve this goal.
The viability of the approach and its robustness to out-of-distribution perturbations are validated with a relatively extensive empirical evaluation, including different ablations of the system (such as one-hot encoding of the features, or no features), variations on the caption lengths or replacing words with synonyms.

**Strengths:**

* S1: Provides extensive empirical evaluation of the proposed system
* S2: The presentation is clear.

**Weaknesses:**

* W1: The LightGBM baseline is underspecified. This baseline is the only one that stands as a reference point that is not an ablated version of the proposed model. However, as I understand it, LightGBM is a framework but not necessarily a model, so I don't really to which model this system is really being compared against.
* W2: Not very clear what is the added value of the proposal of using LLMs against simply using a template-based natural language description.
* W3: Despite the system is motivated on the potential intuitiveness of language interfaces to non-experts, no particular study is conducted on that front.

**Questions:**

* Q1) What's the advantage of the proposed approach using LLMs over more traditional template-based natural language captions? (e.g. "The building is <x> squared feet.", etc.)
* Q2) In Figure 1, the key-value template has only a colon to separate the key and the value. Have you tried adding a space in between? I expect
* Q3) For the one-hot encodings, how do you deal with numeric inputs?
* Q4) In the results in Table 3, why did you expect longer captions to have larger error? I would have had the opposite intuition as shorter captions are more likely to miss important attributes.

---

> ### Author Response · Authors · 2024-11-18
> **Question about Q2**
>
> Dear reviewer Gxzq,
>
> It looks like your question Q2 got cut off - "Q2) In Figure 1, the key-value template has only a colon to separate the key and the value. Have you tried adding a space in between? **I expect...**". Can you provide the rest of this question?
>
> Thank you!

---

> > ### Comment · Reviewer_Gxzq · 2024-11-18
> >
> > Oh, sorry about that! Here's the rest of the question:
> >
> > Q2) In Figure 1, the key-value template has only a colon to separate the key and the value. Have you tried adding a space in between? I expect that some tokenizers could not appropriately segment the key, separator, and value as independent token sequences when they have no space in-between, leading to degraded performance.

---

> ### Author Response · Authors · 2024-11-20
> **Author response**
>
> Dear Reviewer Gxzq,
>
> Thank you for your review!
>
> * **[W1: LightGBM]:** We are using the gradient-boosted decision tree (gbdt) model. We used 1000 estimators. We conducted a grid search over the learning rate, number of leaves, subsampling rate, feature fraction, and min number of data in leaf parameters. The ranges and best values selected based on this hyperparameter sweep are provided in Tables 6 and 7 in the Appendix. We will improve our presentation here in the main text in an updated version of the PDF.
> * **[W2: LLM-generated captions vs. templates]:** Our work considers the use of both a) templated system captions (the colon-separated attribute names and values) and b) “conversational”-style natural language captions generated by an LLM. Some advantages of (b) include: the ability to use the LLM to augment the training data via prompt augmentation (see Section 6.5), the ability to leverage the LLM to invent descriptions for you when trying to come up with a description of a complex system or data (see [1] for a recent example trying to describe a smoke buoyancy simulation), and the potential to use the LLM to rephrase attribute names or values that may appear inscrutable to a non-expert (see our response [W5 - non-experts] to Reviewer 2gzl about this). We highlight that unconstrained, free-form text captions have been used extensively for text-audio and text-music multimodal modeling [2]. This is useful when, for example, a user wishes to search for a song but can only vaguely describe it. We draw an analogy to the application of complex system design—in early stages, an engineer may only have a vague idea about the characteristics that the final system will have. However, as mentioned below [W3], we leave conducting interactive user studies for future work.
> * **[W3 - user evals]:** The reviewer is correct that we did not conduct a study to understand how non-expert users perceive our method. Our empirical evaluation is guided by the question of whether templated captions and LLM-generated captions can achieve good regression performance on real-world systems. The focus of this paper is thus on first establishing the technical feasibility of this general approach.  We argue that previous work does not provide a conclusive answer to this question, and that our work confirms its feasibility.  We will add a discussion in the revised PDF about the importance of conducting user studies to quantify how non-experts perceive language-augmented surrogate models.
>
> * **[Q1]:** See response W2.
> * **[Q2 - tokenization]:**  We did not try adding a space instead of a colon in-between the key and value. We verified that the `bert-base-uncased` tokenizer has no issue tokenizing the colon separately. We provide an example at the bottom of the comment.
> * **[Q3 - numeric vars]:** The building and wind farm simulators have numeric attributes that are “bucketed” by design, i.e., they only take on a fixed number of values. For example, the building square footage attribute only takes on 10 different values. For both datasets, the one-hot encoding baselines all use sklearn’s OneHotEncoder to create one-hot vectors out of all attributes.
> * **[Q4 - short vs. long]:** We expected “long” captions to have larger error because the BERT text encoder is fine-tuned on captions of “medium” length, and Transformers are known to have difficulty with generalizing to longer sequences than seen during training. We could potentially improve generalization from “medium” to “long” captions by, for example, using a text encoder with a more advanced position encoding strategy that is less sensitive to the input sequence length, but we leave this exploration for future work. We will add more qualitative examples of short, medium, and long building captions to the appendix to help convey these intuitions better.
>
> References:
>
> [1] Zhou, Anthony, et al. "Text2PDE: Latent Diffusion Models for Accessible Physics Simulation." arXiv preprint arXiv:2410.01153 (2024).
> [2] Huang, Qingqing, et al. "Mulan: A joint embedding of music audio and natural language." arXiv preprint arXiv:2208.12415 (2022).
>
> Example:
>
> ```python
> >>> x = np.load('10_cap_ids.npy')
> >>> tokenizer.decode(x)
> '[CLS] building _ subtype : none | building _ type : smalloffice | number _ of _ stories : 1. 0 | sqft : 7500. 0 | hvac _ system _ type : psz - ac with no heat | weekday _ operating _ hours : 9. 25 | weekday _ opening _ time : 8. 5 | weekend _ operating _ hours : 8. 5 | weekend _ opening _ time : 6. 75 | tstat _ clg _ delta _ f : 0. 0 | tstat _ clg _ sp _ f : 75. 0 | tstat _ htg _ delta _ f : 6. 0 | tstat _ htg _ sp _ f : 68. 0 [SEP]'
> ```

---

> > ### Author Response · Authors · 2024-11-23
> > **Updated manuscript**
> >
> > Dear Reviewer Gxzq,
> >
> > We have updated the manuscript to reflect the following:
> >
> > * [W1] Specified our use of the GBDT model for LightGBM (Line 370)
> > * [W3] Added “It is natural to expect that non-experts may benefit more from our approach if the LLM is also instructed to simplify the simulator metadata or to provide explanations of technical concepts. Conducting interactive evaluations with non-experts will be important to obtain feedback for further improving the approach.” (Lines 532-535) and “We did not conduct user studies in this work, as we first aimed to establish technical feasibility of this surrogate modeling approach” (Lines 536-538).
> > * [Q4] Added qualitative example comparing the short, medium, and long captions (Appendix, Figure 12). We also provide a new qualitative example for the wind farm experiment showing the various caption style augmentations (Figure 13).

---

> > > ### Comment · Reviewer_Gxzq · 2024-11-25
> > > **Thank you**
> > >
> > > I thank the reviewers for their comprehensive reply to my (and others') review. I have read all the discussions and I feel like the paper has been significantly strengthened by the authors during this discussion by addressing many of the reviewers' concerns.
> > > I don't have any more questions at this point. Still, I would still like to stress that even though evaluating the technical feasibility of the proposed approach is a very valid first step towards the envisioned goals (of free-form text understanding to condition surrogate models), I cannot consider it more than a "fair" contribution in terms of novelty and impact. For this reason, and because the global score scale jumps from 6 directly to 8 (which has the very poor label "accept, good paper"), I don't think I can improve the scores any further. Just to be clear, I do think this is a "good paper" (which is why I find the labelling of the scale so lacking), but I cannot justify assigning it a score of 8, as that would imply a stronger endorsement than I am prepared to give. My evaluation remains that the paper makes a meaningful contribution and is worthy of publication, but it does not stand out as particularly innovative or high-impact to warrant a higher score. I hope this feedback helps clarify my position, and I thank the authors again for their diligent efforts in addressing the reviewers’ concerns.

---

> > > > ### Author Response · Authors · 2024-11-25
> > > > **Thanks**
> > > >
> > > > Thank you for contributing your time to reviewing and discussing our paper. We are glad to hear that your assessment is that our paper  **"makes a meaningful contribution and is worthy of publication"**. We understand and sympathize with the limitations of the score scale and your hesitation to increase your score.
> > > >
> > > > However, we kindly ask you to raise your score, since, after discussion, the assessment is that the paper makes a meaningful contribution and is worthy of publication. Due to the limited score scale, a paper that is "particularly innovative and high-impact" would likely receive a score *higher* than 8. The act of raising your score post-rebuttal would also help confirm that "the paper has been significantly strengthened by the authors during this discussion by addressing many of the reviewers' concern".
> > > >
> > > > Again, we thank you for taking the time to read our paper and engage in discussions!

---

> > > > > ### Comment · Reviewer_Gxzq · 2024-11-26
> > > > >
> > > > > I accept the authors arguments and so, while I still keep my score for the level of contribution, I have now changed my global assessment to 8 at the risk of erring on the positive rather than on the negative side.

---

> > > > > > ### Author Response · Authors · 2024-11-27
> > > > > > **Thank you**
> > > > > >
> > > > > > Dear Reviewer Gxzq,
> > > > > >
> > > > > > Thank you for increasing the rating to 8. We sincerely appreciate the time and effort you have dedicated to reviewing our paper and engaging in discussions.
> > > > > >
> > > > > > -Authors

---

### Official Review · Reviewer_2gzL · 2024-11-04

**Soundness:** 3
**Presentation:** 3
**Contribution:** 2
**Rating:** 8
**Confidence:** 3

**Summary:**

The paper discusses the important challenge of building surrogate models for the prediction of simulation data. They specifically motivate the problem for complex energy systems(CES). These surrogate models often model system features as one hot vectors. The authors propose using text based descriptions to model these so-called surrogate systems with time series data. The text data is encoded as a dense embedding obtained from language models. The embedding is then fed to a bidirectional sequence encoder along with the time series data.

The paper discusses the generation of the text pertaining to the attributes of such systems and proposes an automatic evaluation strategy for the same.

For generating the captions the authors prompt an LLM with an in-context learning-based prompt that tunes the style and number of sentences. To evaluate the SysCap quality the authors train a multi-class classifier to check the attributes covered in the description generated by the LLM, using the text embedding.

The authors show how including SysCaps along with time series data leads to improved performance against baselines that perform onehot encoding over attributes. The authors further show how training a custom embedding model can aid in improving time series prediction over just using a time series-based model. They further empirically prove how the embeddings are more robust to synonyms and missing data.

**Strengths:**

The paper is well-written and easy to follow. The authors motivate the problem well and empirically show improvements over 2 real-world datasets. Further, SysCaps can be used by non-expert users to understand the features of surrogate systems. The Design space exploration is insightful to show the features learned by the model.

**Weaknesses:**

1. The paper claims the technique uses a pretrained text encoder for generating the embeddings, but then in section 5 mentions that the models are actually finetuned. This should be explicitly mentioned in the claims that the paper makes rather than just mentioning that a pretrained embedding is used.
2. Further, the authors do not compare with the "said-pretrained" embeddings but only finetuned embeddings, and other SOTA embedding models for text encoding.
3. The paper also claims that they train a system to evaluate the caption quality, the parameters of the said multiclass classifier are omitted from the paper.
4. The paper claims that for the CES building energy consumption dataset, the SysCaps-kv configuration works best, and for the turbine configuration the SysCaps-nl, there should be some discussion regarding the insights drawn from both cases and why the performance for both techniques are different.
5. The authors claim that SysCaps would be useful for non-expert users, but lack the discussion if LLM-based explanations (complementary to the work done) can also aid in explaining the system attributes for surrogate models.

**Questions:**

In addition to the points in the weakness:

1. Did the authors try to just templatize the sentences rather than generating them using an LLM, how would that impact performance (i.e. rather than telling an LLM to adhere to some constraint-based template, just have a sketch sentence and fill attribute values in the given sentence)?
2. Why wasn't RFE performed for the Wind Farm Wake modeling dataset, would performing RFE improve performance ?
3. Would the model not further improve if the SysCaps were generated using synonyms for the attributes, did the authors see the LLM generate different synonyms for the building or wind farm dataset?
4. Do the authors believe that training on the subset of data where the caption quality assessed by the classifier model, would improve the overall model performance?

---

> ### Author Response · Authors · 2024-11-20
> **Author response**
>
> Dear Reviewer 2gzl,
>
> Thank you for your review and for praising the presentation, motivation, and empirical validation of the work.
>
> * **[W1,W2 - pretrained vs. finetuned embeddings]:** We will clarify in our Introduction that we propose to *fine-tune* text embedding models initialized from pretrained weights. We are working on adding a new ablation to the PDF that shows the performance of our SysCaps-nl model without fine-tuning the BERT text encoder (i.e., only using the pretrained embeddings). We are also going to run an ablation without a "SOTA embedding model"---“SimCSE-RoBERTa-large” , which is the current most-downloaded text encoder on HuggingFace as of 11/19/24.
> * **[W3 - classifier hyperparams]:** We will add the hyperparameter details for the multiclass classifier to the appendix. We implement the classifier on top of the text encoder by adding a linear layer for each attribute type, where this layer predicts logits for each attribute's  classes.  We use AdamW with lr=3e-4, early stopping with patience 5, batch size 128, and max epochs 100. We do not freeze the text encoder weights.
> * **[W4 - kv vs. nl]:** On the comparison between SysCaps-kv on the building dataset and SysCaps-nl on the wind dataset, we believe the relative performance difference is easily explained. The wind dataset, which only has 500 total systems (300 training, 100 validation, 100 testing), the SysCaps-nl model uses prompt augmentation to increase both the quantity and quality (e.g., higher variability in syntax and style) of the training data. On the buildings dataset, SysCaps-nl does not use prompt augmentation. This is because the size of the buildings dataset was too large to create multiple captions per building. We expect that training the SysCaps-nl model on the wind dataset *without* prompt augmentation would result in similar (or worse) performance than SysCaps-kv. We will run this experiment and add the results.
> * **[W5 - non-experts]:** We agree that our method has the potential to help explain system attributes for surrogate models to non-experts. The current manuscript lacks discussion about this, which we will rectify. We emphasize, however, that evaluating this goes beyond the scope of our current methodology. For example, we imagine this would require instructing the LLM to identify system attributes whose names might be inscrutable to humans, and to then use the rest of the metadata and system prompt to come up with better names that are easier for non-experts to use. Then, the LLM could be instructed to use these new names to create the SysCaps. We will highlight this when discussing future work and potential impacts in our Conclusion.
>
>
> * **[Q1 - Template sentences]:** We did not try training with template sentences. However, we consider the “key-value” captions used by the SysCaps-kv models, which are lists of attribute names and value `key1:value1 | key2:value2 | …` as an estimate for the performance of sentence templates. We believe this is a reasonable assumption because a sentence template just adds extra words which do not possess new information about the system being simulated.
> * **[Q2 - RFE]:** There are only 5 attributes for the wind dataset, which is relatively few and so RFE is not needed. The captions generated by the LLM for the wind dataset are also relatively short.
> * **[Q3 - augmenting training]** Based on the results of our prompt augmentation experiment (Section 6.5), we have evidence that, yes, increasing both the quantity and quality of the training data via augmentation would lead to better performance. This can include generating extra captions by prompting the LLM to create captions using attribute synonyms. We will add more qualitative examples of the LLM-generated captions to the appendix in the revised PDF. Here is an example where the LLM paraphrased (bolded):
>   * "The wind plant is designed with a cluster layout, featuring 40 turbines of varying heights. Each turbine boasts a rotor diameter of 130 meters, providing ample sweep area to capture the wind's energy. With a **mean turbine spacing** of 7 times the rotor diameter, the plant is optimized for efficient energy production, while minimizing visual impact and land usage.”
>   * “The wind plant features a layout of multiple strings, with 73 turbines standing tall at an impressive rotor diameter of 130 meters. The **turbines are spaced at an average distance** of four times the rotor diameter, resulting in a highly efficient and productive wind farm. Each turbine is equipped with a rated power of 3.4 megawatts, making it capable of generating a significant amount of electricity from the wind.”
>
> For Question 4, perhaps you can clarify—is the question about whether we could improve performance by using a threshold to filter out “low quality” captions?

---

> > ### Comment · Reviewer_2gzL · 2024-11-23
> >
> > Clarification for question 4: yes I was trying to reason if using a threshold to filter lower-quality captions. Recent work by [Singh et. al](https://arxiv.org/abs/2407.10657). shows that synthetically generated LLM captions harm smaller open-source LLM models more than larger LLMs. In the given case, since the authors identify lower-quality captions using some criteria, it would be insightful to see the performance of the high-quality subset.

---

> > > ### Author Response · Authors · 2024-11-23
> > > **Reply, updated manuscript**
> > >
> > > Dear Reviewer 2gzl,
> > >
> > > Thanks for the clarification for question 4. In the paper you shared, there is a binary notion of correctness for the LLM-generated caption, which is used to filter out low-quality captions for fine-tuning. Our setting is slightly different and more complex---we train *multi-class classifiers* to assess the correctness of the captions holistically in terms of all system attributes. While we agree that it is interesting to ask whether performance can be improved by training on only high-quality captions, we believe that determining how to do this properly for our setting deserves careful consideration and the experimentation involved will take more time than what the rebuttal period allows for.
> > >
> > > We have updated the manuscript with the following:
> > >
> > > * [W1] Clarified in the introduction that we fine-tune the language embeddings (Line 69).
> > > * [W2] We trained SysCaps-nl with BERT without fine-tuning (pretrained only), and the results were poor (Stock-Annual NRMSE = 0.356) (Lines 419-420). We also tried training a version of our best buildings model, SysCaps-kv, using RoBERTa-large (sup-simcse-roberta-large), a text encoder which is more similar to recent "SOTA" text embedding approaches. The results are unsurprisingly comparable to BERT. `Sup-SimCSE RoBERTa-Large - Buildings-Hourly NRMSE = 0.488`  vs. `BERT - Buildings-Hourly NRMSE = 0.450` and `Sup-SimCSE RoBERTa-Large - Stock-Annual NRMSE = 0.027` vs. `BERT - Stock-Annual NRMSE = 0.020`.
> > > * [W3] Added multi-class classifier hyper-parameters to the appendix (Lines 956-958).
> > > * [W4] We trained a SysCaps-nl model without prompt augmentation on the wind farm dataset and added this result to the PDF. The performance slightly decreases, as expected (from an NRMSE of 0.035 with prompt augmentation to an NRMSE of 0.038 without augmentation). Note that before, the NRMSE of the SysCaps-nl model was 0.036, but we discovered that this was computed with prompt augmentation still enabled.
> > > * [W5] We added “It is natural to expect that non-experts may benefit more from our approach if the LLM is also instructed to simplify the simulator metadata or to provide explanations of technical concepts. Conducting interactive evaluations with non-experts will be important to obtain feedback for further improving the approach.” (Lines 532-535) and “We did not conduct user studies in this work, as we first aimed to establish technical feasibility of this surrogate modeling approach” (Lines 536-538).
> > > * [Q3]  We added more qualitative examples to the appendix (Appendix Figures 12 and 13) that visualize the diversity in text produced by the LLM when creating synthetic SysCaps, along with the model predictions, for specific building and wind farm test examples.

---

> > > > ### Comment · Reviewer_2gzL · 2024-11-25
> > > >
> > > > I thank the authors for their clarifications and newer ablations conducted during the rebuttal phase. I have increased my score to reflect this.
> > > >
> > > > Overall, it is interesting to see the use of LLMs in modeling systems that can model captions and take account of time series data in a multimodal manner. Future improvements in this line of work would be useful.

---

> > > > > ### Author Response · Authors · 2024-11-26
> > > > > **Thank you**
> > > > >
> > > > > Dear Reviewer 2gzL,
> > > > >
> > > > > Thank you for increasing the rating to 8. We are glad that our response has addressed your questions. We would be happy to assist if you have any follow-up questions or wish to discuss further.
> > > > >
> > > > > We appreciate the time and effort you have dedicated to reviewing our paper and engaging in discussions.
> > > > >
> > > > > -Authors

---

### Official Review · Reviewer_38tw · 2024-11-05

**Soundness:** 2
**Presentation:** 3
**Contribution:** 2
**Rating:** 5
**Confidence:** 2

**Summary:**

This paper describes a set of lightweight models to model complex energy systems, using an LLM to generate prompts and a encoder and bidirectional time-series model to predict energy consumption.

**Strengths:**

- The authors provide a clear explanation of their setup, flow of data across multiple components and their evaluation and analysis.
- (I do not feel sufficiently well-acquainted with this domain to evaluate the predictive contribution or performance of the models.)

**Weaknesses:**

- I presume the authors' choice of models is due to resource constraints and aiming for a lightweight setup, but it feels like it has multiple components when it could be a simpler setup with fewer model components. For instance, a BERT-type model could also be used for time-series prediction (as opposed to only text encoding). Similarly the two-step process of generating prompts using a separate LLM and then encoding that prompt with an encoder could be avoided by just using the LLM directly and fine-tuning it.

**Questions:**

I would be interested in seeing the performance compared between time-series-centric models and current generic architectures.

---

> ### Author Response · Authors · 2024-11-20
> **Author response**
>
> Dear Reviewer 38tw,
>
> Thank you for your review! A single fine-tuned language-model (LM) for timeseries regression would be appealing. However, we can justify our choice to instead use two separate components for encoding text and performing timeseries regression.
>
> *  **[Limited context window size of LMs]:** A single simulation for a complex system may run for thousands of time steps. For example, the building surrogate models in our experiments predict hourly energy consumption for 1 year (T= 8,760). This hinders the use of language models—BERT has a context window size of 512, and even alternative models such as LongFormer and Llama-2-7b have a max context window size of 4,096. Of course, there exist advanced tricks to aggregate multiple embeddings computed for a single document where the document length exceeds the context window. We believe the RNNs and State Space Models (SSMs) we use offer a simpler solution as they have no such limitations on the sequence length.
> * **[Computational expense of LMs]:** The quadratic complexity of the self-attention mechanism in Transformer-based language models, as well as the high number of parameters in LLMs such as Llama-2-7b, makes LMs poor choices for surrogate models of complex systems in practice. For a concrete example, to conduct the sensitivity analysis case study (Figure 4), we perform 960K model predictions, which took us ~1 hour on 1 NVIDIA A100-40GB GPU. If we generously assume that Llama-2-7b takes 10 seconds to generate 8,760 tokens (this is assuming a generation speed of 876 tokens/sec), this case study would have taken 9.6M seconds, or ~111 days, to run. Our multi-component approach helps make our framework ready to be used by practitioners.
> * **[Our two-step process vs. one-step]:** In our framework, at train time the first step is to use an LLM to generate synthetic natural language SysCaps. The second step is to encode the text caption for multimodal timeseries regression. The first step is only used at training time---at test time, a person can interact with the multimodal surrogate model and prompt it with a text description. While the idea of using a single LLM to directly perform timeseries regression is intuitive, as previously stated, we intentionally want to avoid using an LLM as a text encoder due to the computational burden. Moreover, an additional advantage of our approach is in its modularity. We can decide to swap out the Llama-2-7b LLM for a more powerful LLM such as Claude or GPT-4 to generate training captions. This is not so easy if the LLM is also fine-tuned for timeseries regression. Likewise, we can swap out the lightweight text encoder model for a more powerful one. Our experiments comparing DistillBERT and BERT (Table 1) show that larger, more powerful text encoders lead to better regression accuracy.
>
> For the question about a comparison between time-series centric models and generic architectures, could you give us a specific model that you have in mind? The exact comparison you are requesting is unclear to us.

---

> > ### Author Response · Authors · 2024-11-23
> > **Question for reviewer**
> >
> > Thanks again for your review. We have a question about one of your comments, which we are re-posting in case it got lost in our response above:
> >
> > With regards to the requested comparison between time-series centric models and generic architectures, could you give us a specific model that you have in mind? The exact comparison you are requesting is unclear to us.

---

### Meta-Review · Area_Chair_ActA · 2024-12-24

**Metareview:**

Reviewers liked that the paper solves the well-motivated problem of interacting with complex energy system (CES) using natural language and the strength of the results. They did not evaluate much of the technical aspects and provided lower confidence reviews: 38tw: could use BERT to directly predict the time series. 2gzL: unclear if fine-tuned or not. Gxzq: how LightGBM was used as baseline is not clear. It is clear that the paper focuses on specialized area that could be important, but also very few reviewers understand. This is also out of the domain for me, so I defer to reviewers on the importance of the problem and recommend accept to err on the positive side.

**Additional Comments On Reviewer Discussion:**

Not a typical ML paper and the system did not match to the correct area if there are enough people in the area.
The review completion was was low, with only 3/6 reviewers submitting.

---

### Decision · Program_Chairs · 2025-01-22

Accept (Poster)